# Nutritional Biomarkers and Heart Rate Variability in Patients with Subacute Stroke

**DOI:** 10.3390/nu14245320

**Published:** 2022-12-15

**Authors:** Eo Jin Park, Seung Don Yoo

**Affiliations:** Department of Rehabilitation Medicine, Kyung Hee University Hospital at Gangdong, Seoul 05278, Republic of Korea

**Keywords:** nutrition biomarkers, malnutrition, heart rate variability, autonomic dysfunction, stroke

## Abstract

Malnutrition and autonomic dysfunction are associated with poor outcomes, mortality, and psychological problems after stroke. Relevant laboratory biomarkers include serum albumin, prealbumin, and transferrin. Heart rate variability (HRV), a noninvasive measurement, can objectively measure autonomic nervous system (ANS) function. The relationship between HRV and nutritional biomarkers in stroke patients has not been studied. This study aimed to examine the relationship between nutritional biomarkers and HRV parameters in stroke patients. We retrospectively recruited 426 patients with subacute stroke who were examined for nutritional biomarkers, such as serum albumin, prealbumin, and transferrin, and underwent 24 h ambulatory Holter electrocardiography. Patients were divided into groups according to their nutritional biomarker status. Differences in HRV parameters between nutritional biomarker-deficient and normal groups were assessed. Pearson’s correlation and multiple regression analyses were used to verify the relationship between HRV parameters and nutritional biomarkers. HRV parameters were significantly lower in the nutritional biomarker-deficient groups. In addition, there was a significant association between HRV parameters and nutritional biomarkers. Serum albumin, prealbumin, and transferrin levels were associated with ANS function, as measured by HRV, and their deficiency may be a predictive factor for the severity of ANS dysfunction in stroke patients.

## 1. Introduction

Malnutrition is a major problem in stroke patients. Although malnutrition in stroke patients is underdiagnosed and undertreated, its prevalence at admission is predicted to range from 6.1% to 62% [1]. Poor nutritional status leads to poor prognoses, especially after a stroke and has been linked to worse functional outcomes, longer hospital stays, and increased mortality risk [2,3,4]. Malnutrition assessment is complex and requires a multidisciplinary approach. Relevant laboratory biomarkers that are easy to measure and widely used include serum albumin, prealbumin, and transferrin [5]. Several biomarkers have been identified to evaluate malnutrition. When blood albumin levels are low, muscle mass diminishes [6,7]. Hypoalbuminemia is used to predict the mortality of older individuals living in the community, nursing homes, or hospitals. Hypoalbuminemia is affected by inflammation and is characterized by high levels of TNF-α and IL-6. Prealbumin is produced by the liver and is partially catabolized by the kidneys. Low levels of serum prealbumin are correlated with malnutrition [8]. Serum prealbumin levels are positively associated with decreased muscle mass in the elderly. During states of protein deficiency, the liver creates less prealbumin [6]. When prealbumin levels decrease, they may be utilized to determine the severity of involutional lean body mass processes by discriminating between hepatic and muscle abnormalities caused by protein restriction [9]. Transferrin may also function as a nutritional status indicator [5]. Transferrin is regarded as a useful indicator of nutritional health [10]. Serum transferrin levels decrease in the presence of severe malnutrition. However, this marker is unable to reliably measure mild malnutrition in the elderly [11].

The heart rate variability (HRV) has been proven to reflect the autonomic nervous system (ANS) activity, which regulates almost all vascular, visceral, and metabolic activities [12,13]. HRV abnormalities indicate autonomic imbalance and are associated with poor cardiovascular prognoses [14]. Low HRV is an independent risk factor for cardiovascular mortality [15,16]. Several studies on stroke patients have shown significant evidence that HRV parameters may act as biomarkers for incident stroke and a variety of post-stroke outcomes, including functional prognosis and motor function [17,18]. Abnormalities in frequency-domain HRV parameters, including low frequency (LF) and high frequency (HF), are associated with post-stroke infection, activities of daily living, and functional outcomes [17]. In addition, time-domain HRV parameters, such as root mean square difference between successive R-R intervals (rMSSD) and standard deviation of all normal beat interval (SDNN), are associated with stroke severity, motor impairment, mortality, and functional outcome [17,19].

HRV is a noninvasive measurement that can objectively measure ANS function. A decrease in HRV indicates ANS dysfunction. In particularly, the LF and LF/HF ratios reflect sympathetic tone, and HF reflects parasympathetic tone [17,20]. The relationship between HRV and nutritional biomarkers in stroke patients has not been studied. This study aimed to examine the relationship between nutritional biomarkers and HRV parameters in patients with stroke.

## 2. Materials and Methods

### 2.1. Subjects

Subacute stroke patients hospitalized at the Kyung Hee University Hospital in Gangdong from September 2010 to April 2022 were included in this retrospective study. Among the patients, we recruited those who were examined for nutritional biomarkers such as serum albumin, prealbumin, and transferrin, and underwent 24 h ambulatory Holter electrocardiography. Patients with first-onset stroke and those who underwent testing within three months of onset were included. Patients with a history of diseases, such as liver, inflammatory bowel, thyroid, and metabolic disease and nephrotic syndrome, that could affect albumin, prealbumin, and transferrin levels were excluded. To investigate whether there was a difference in HRV parameters according to albumin, prealbumin, and transferrin deficiency, patients were divided into deficiency and normal groups. The study was performed according to a protocol authorized by the Institutional Review Board of Kyung Hee University Hospital in Gangdong, Korea (IRB approval number: 2022-10-048).

### 2.2. Nutritional Biomarkers

Serum albumin, prealbumin, and transferrin levels were measured by blood sampling. The cut-offs for albumin, prealbumin, and transferrin deficiency were ≤3.5 g/dL [21], ≤20 mg/dL [22], and ≤200 mg/dL [23], respectively.

### 2.3. Heart Rate Variability

The HRV was evaluated using 24 h ambulatory Holter electrocardiography with three channels (GE Healthcare, Milwaukee, WI, USA). Prior to the recording, each patient was placed in a supine position for at least 10 min in a silent environment. The digitized data were recorded at a sampling frequency of 128 Hz. The R-R interval, interval between consecutive heartbeats, and QRS complex were measured by a single examiner using the same electromyography equipment. Time-domain features included SDNN in milliseconds, mean of five-minute standard deviations of intervals (ASDNN), standard deviation of five-minute mean R-R intervals (SDANN), percentage of intervals that were more than 50 ms different from the prior interval (pNN50), and rMSSD. The frequency-domain features included very low frequency (VLF; 0.003–0.04 Hz), HF (0.15–0.40 Hz), LF (0.04–0.15 Hz), and the LF/HF ratio utilizing standard fast Fourier transformation.

### 2.4. Statistical Analysis 

SPSS version 20.0 for Windows (IBM Corp., Armonk, NY, USA) was used to conduct statistical analyses. The Kolmogorov–Smirnov test was performed to determine the normality of the data distribution, and Levene’s test was used to check the homogeneity of variance. To compare HRV parameters between subgroups, an independent t-test was conducted. The correlations between serum albumin, prealbumin, and transferrin levels and HRV parameters were examined using Pearson’s correlation coefficient. The influence of serum albumin, prealbumin, and transferrin levels on HRV parameters was examined using multiple linear regression analysis with stepwise adjustment for age, sex, type of stroke, comorbidities, medication, MMSE (Mini-Mental State Examination), MBI (modified Barthel index), and BMI (body mass index). In all statistical tests, a *p*-value < 0.05 was regarded as statistically significant.

## 3. Results

### 3.1. Baseline Characteristics

A total of 426 patients were recruited with a mean age of 67.03 ± 13.03 years. The study included 197 men and 229 women. The classification according to stroke type included 278 and 148 patients with ischemic stroke, respectively. Baseline characteristics, comorbidities, medications, nutritional biomarkers, and HRV parameters are shown in Table 1.

### 3.2. Comparison of the HRV Parameters among Subgroups Categorized by Serum Albumin Level

In the frequency domain of HRV parameters, the VLF, LF, HF, and LF/HF ratio were significantly lower in the albumin-deficient group. In the time domain, the albumin-deficient group showed significantly lower values for SDNN, ASDNN, and rMSSD. Although the SDANN and pNN50 values were lower in the albumin-deficient group, the difference was not statistically significant (Table 2).

### 3.3. Comparison of the HRV Parameters among Subgroups Categorized by Serum Prealbumin Level

In the frequency domain of the HRV parameters, the VLF, LF, HF, and LF/HF ratio were significantly lower in the prealbumin-deficient group. In the time domain, SDNN and ASDNN had significantly lower values in the prealbumin-deficient group. The prealbumin-deficient group showed lower values for SDANN, rMSSD, and pNN50; however, the difference was not statistically significant (Table 3).

### 3.4. Comparison of the HRV Parameters among Subgroups Categorized by Serum Transferrin Level

In the frequency domain of the HRV parameters, the VLF, LF, HF, and LF/HF ratio were significantly lower in the transferrin-deficient group. In the time domain, SDNN, SDANN, ASDNN, rMSSD, and pNN50 showed lower values in the transferrin-deficient group, but the differences were not statistically significant (Table 4).

### 3.5. Correlation of Serum Albumin, Prealbumin, Transferrin Level with HRV Parameters

A statistically significant correlation was reported between serum albumin levels and all frequency domains of the HRV parameters (VLF: *r* = 0.312; LF: *r* = −0.223; HF: *r* = 0.218; LF/HF ratio: *r* = 0.256). In the time domain, there was a statistically significant correlation with SDNN (*r* = 0.124), SDANN (*r* = 0.099), ASDNN (*r* = 0.116), and rMSSD (*r* = 0.122), but not with pNN50. There was a statistically significant correlation between serum prealbumin levels and all frequency domains of the HRV parameters (VLF: *r* = 0.313; LF: *r* = 0.226; HF: *r* = 0.257; LF/HF ratio: *r* = 0.257). In the time domain, there was a significant correlation with SDNN (*r* = 0.127), SDANN (*r* = 0.116), ASDNN (*r* = 0.122), and rMSSD (*r* = 0.127), but not with pNN50. There was a statistically significant correlation between serum transferrin levels and all frequency domains of the HRV parameters (VLF: *r* = 0.307; LF: *r* = 0.219; HF: *r* = 0.254; LF/HF ratio: *r* = 0.254). In the time domain, there was a significant correlation with SDNN (*r* = 0.126), SDANN (*r* = 0.122), ASDNN (*r* = 0.121), and rMSSD (*r* = 0.121), but not with pNN50 (Table 5).

In the multiple regression analysis, serum albumin levels (R^2^ = 0.174) measured in VLF (β = 0.005, *p* < 0.001), LF/HF ratio (β = 0.32, *p* = 0.001), LF (β = 0.005, *p* = 0.002) and HF (β = 0.010, *p* = 0.005) were significant predictors. Serum prealbumin levels (R^2^ = 0.176) measured in the VLF (β = 0.003, *p* < 0.001), LF/HF ratio (β = 0.207, *p* = 0.001), LF (β = 0.003, *p* = 0.002), and HF (β = 0.006, *p* = 0.005) were significant predictors. Serum transferrin levels (R^2^ = 0.170) measured in VLF (β = 0.089, *p* < 0.001), LF/HF ratio (β = 0.485, *p* = 0.001), LF (β = 0.081, *p* = 0.002), and HF (β = 0.171, *p* = 0.005) were significant predictors (Table 6).

## 4. Discussion

To the best of our knowledge, this is the first study to evaluate the correlation between nutritional biomarkers and HRV measures that indicate ANS function in patients with subacute stroke. In this study, serum albumin, prealbumin, and transfer levels, which are used as nutritional biomarkers to evaluate nutritional status, were correlated with HRV parameters measured by 24 h ambulatory Holter electrocardiography. The albumin-deficient and prealbumin-deficient groups had lower HRV parameters in all frequency domains and in some time domains as compared to the normal group. The transferrin-deficient group had lower HRV parameters in all frequency domains as com-pared to the normal group. These findings suggest that these nutritional biomarkers are clinically significant for predicting ANS function, as evaluated using HRV.

Since acute stroke often generates changes in cardiovascular interactions, post-stroke blood pressure and heart rate upon admission might be possible indicators of individuals at risk of poor prognosis [24,25]. However, it is difficult to measure the effect on the ANS simply by measuring blood pressure and heart rate. Complex data are contained within the heart rhythm, showing the indicative of several overlapping systems. ANS dysfunction is prevalent in patients with stroke [26,27]. ANS dysfunction is characterized by a decrease in HRV, an increase in sympathetic tone as measured by LF and the LF/HF ratio, and a decrease in parasympathetic tone as measured by HF [17]. They are caused by injury to the central autonomic network, namely, in the frontoparietal cortex and brainstem, or disruption of autonomic routes from the hypothalamus to the midbrain, pons, and spinal cord [28]. The most prevalent clinical disorders are anomalies in heart rate and blood pressure control, which represent cardiovascular autonomic dysfunction, and sweating reflect alterations in the vasomotor and sudomotor regulatory systems [27]. Moreover, autonomic dysfunction after stroke affects the respiratory and sexual systems [29]. In the early period of stroke, cardiovascular autonomic dysfunction, which is mostly due to increased sympathetic activity, is most apparent; however, other autonomic problems, such as excessive sweating, are chronic or even permanent [27]. In addition to the well-known sympathetic malfunction, parasympathetic nervous system anomalies may also contribute to autonomic dysfunction following stroke [26,30]. The use of quantitative analytical techniques to identify autonomic dysfunction reliably is crucial. In addition, quantitative data serves as the basis for the sequential treatment of a variety of autonomic diseases associated with stroke [28].

Several studies have reported an association between malnutrition and autonomic dysfunctions. Serum albumin, prealbumin and transferrin have been reported to be related to body mass index and hydration status [31,32,33]. Low body mass index and fluid hydration status were independently associated with elevated sympathetic indices such as LF and the LF/HF ratio [34,35]. Other studies have also reported an association between serum albumin, prealbumin, and body fat mass [36,37]. rMSSD is an indicator of the parasympathetic activity index and has a negative correlation with the hip-to-waist ratio and body fat mass [38]. An increase in lean body mass and body weight is associated with improved survival in hemodialysis patients, whereas a decrease in body weight is associated with increased mortality [39]. This is due to the fact that the negative impact of malnutrition partially balances out the alleged dangers of a higher body mass index until the accumulated cardiovascular risks outweigh its preventive effect [40]. ANS dysfunction may begin with hyperactivity of sympathetic tone and a decrease in parasympathetic tone, resulting in sympathetic withdrawal during stressful situations, but typically manifests as overall decrease in HRV [34]. In the multiple linear regression analysis of this study, the decrease in VLF reflecting thermal control of vasomotor tone, the decrease in LF/HF ratio and LF reflecting sympathetic tone, the decrease in LF, and the decrease in HF reflecting parasympathetic tone, were significantly associated with low nutritional [34,41]. A poor nutritional status occurs due to the interaction of multiple factors in stroke patients and results in severe morbidity and mortality. Our findings reported a correlation between malnutrition and autonomic dysfunction, possibly leading to a negative prognosis in individuals with subacute stroke.

This study had several limitations. First, this was a retrospective, cross-sectional study. Second, patients with ischemic and hemorrhagic stroke were recruited. Finally, it is not known whether malnutrition persisted long before onset. Large-scale prospective studies are needed in the future.

## 5. Conclusions

In conclusion, a poor nutritional status, as measured by nutritional biomarkers, such as serum albumin, prealbumin, and transferrin levels, was associated with ANS dysfunction, as measured by HRV parameters. The ANS dysfunction may be an evidence of a correlation between poor nutritional status and stroke.

## Figures and Tables

**Table 1 nutrients-14-05320-t001:** Clinical characteristics of patients.

Characteristic	Value
Age (years)	67.03 ± 13.03
Sex	
Male	197 (46.20)
Female	229 (53.80)
Type of stroke	
Ischemic	278 (65.30)
Hemorrhagic	148 (34.70)
Comorbidities	
Hypertension	260 (61.00)
Arrhythmia	74 (17.40)
Diabetes mellitus	158 (37.10)
Dyslipidemia	325 (76.30)
Coronary artery disease	22 (5.20)
Heart failure	70 (16.40)
Medication	
Beta blocker	87 (20.40)
Calcium channel blocker	193 (45.30)
ACE-i/ARB	199 (46.70)
Diuretics	228 (53.50)
MMSE	14.08 ± 8.25
MBI	39.35 ± 17.62
BMI (kg/m^2^)	21.66 ± 2.06
Nutritional biomarkers	
Albumin (g/dL)	2.99 ± 0.58
Prealbumin (mg/dL)	19.36 ± 3.72
Transferrin (mg/dL)	246.77 ± 99.62
HRV parameter (Frequency domain)	
VLF (ms^2^)	619.06 ± 223.28
LF (ms^2^)	421.44 ± 154.90
HF (ms^2^)	179.04 ± 71.17
LF/HF ratio	5.55 ± 2.66
HRV parameter (Time domain)	
SDNN (ms)	100.51 ± 28.52
SDANN (ms)	101.87 ± 29.10
ASDNN (ms)	50.03 ± 11.74
rMSSD (ms)	26.34 ± 6.23
pNN50 (%)	49.41 ± 23.60

Values are presented as the mean ± standard deviation or number (%). ACE-I, angiotensin-converting enzyme inhibitors; ARB, angiotensin II receptor blockers; MMSE, Mini-Mental State Examination; MBI, modified Barthel index; BMI, body mass index; HRV, heart rate variability; VLF, very low frequency; LF, low frequency; HF, high frequency; SDNN, standard deviation of intervals of all normal beats; SDANN, standard deviation of five-minute mean R-R interval; ASDNN, mean of five-minute standard deviations of intervals; rMSSD, root mean square of the difference of successive R-R intervals; pNN50, percentage of intervals that were more than 50 ms different from the previous interval.

**Table 2 nutrients-14-05320-t002:** Comparison of the HRV parameters among subgroups categorized by serum albumin level.

	Albumin-Deficient Group (n = 321)	Normal Group (n = 105)	*p*-Value
Frequency domain			
VLF (ms^2^)	543.89 ± 264.70	602.81 ± 230.75	0.029 *
LF(ms^2^)	376.53 ± 180.21	424.84 ± 163.12	0.011 *
HF(ms^2^)	147.72 ± 82.09	183.20 ± 71.82	0.003 *
LF/HF ratio	4.13 ± 2.90	5.70 ± 2.58	0.003 *
Time domain			
SDNN (ms)	90.53 ± 31.02	99.32 ± 30.41	0.01 *
SDANN (ms)	92.61 ± 34.12	99.32 ± 30.41	0.58
ASDNN (ms)	47.44 ± 13.27	50.29 ± 11.48	0.03 *
rMSSD (ms)	24.90 ± 7.43	26.70 ± 6.69	0.02 *
pNN50 (%)	45.75 ± 27.47	49.85 ± 23.00	0.13

Values are presented as the mean ± standard deviation. * *p* < 0.05. HRV, heart rate variability; VLF, very low frequency; LF, low frequency; HF, high frequency; SDNN, standard deviation of intervals of all normal beats; SDANN, standard deviation of five-minute mean R-R interval; ASDNN, mean of five-minute standard deviations of intervals; rMSSD, root mean square of the difference of successive R-R intervals; pNN50, percentage of intervals that were more than 50 ms different from the previous interval.

**Table 3 nutrients-14-05320-t003:** Comparison of the HRV parameters among subgroups categorized by serum prealbumin level.

	Prealbumin-Deficient Group (n = 237)	Normal Group (n = 189)	*p*-Value
Frequency domain			
VLF (ms^2^)	585.99 ± 250.44	664.85 ± 260.34	0.002 *
LF(ms^2^)	391.12 ± 175.31	430.32 ± 161.25	0.017 *
HF(ms^2^)	159.54 ± 79.64	186.34 ± 70.81	0.003 *
LF/HF ratio	4.98 ± 2.87	5.58 ± 2.62	0.026 *
Time domain			
SDNN (ms)	93.46 ± 30.39	99.85 ± 29.03	0.028 *
SDANN (ms)	95.34 ± 32.05	99.51 ± 30.91	0.174
ASDNN (ms)	48.01 ± 12.56	50.85 ± 11.39	0.015 *
rMSSD (ms)	25.85 ± 6.96	26.57 ± 6.87	0.286
pNN50 (%)	48.75 ± 25.97	48.91 ± 22.75	0.946

Values are presented as the mean ± standard deviation. * *p* < 0.05. HRV, heart rate variability; VLF, very low frequency; LF, low frequency; HF, high frequency; SDNN, standard deviation of intervals of all normal beats; SDANN, standard deviation of five-minute mean R-R interval; ASDNN, mean of five-minute standard deviations of intervals; rMSSD, root mean square of the difference of successive R-R intervals; pNN50, percentage of intervals that were more than 50 ms different from the previous interval.

**Table 4 nutrients-14-05320-t004:** Comparison of the HRV parameters among subgroups categorized by serum transferrin level.

	Transferrin-Deficient Group (n = 161)	Normal Group (n = 265)	*p*-Value
Frequency domain			
VLF (ms^2^)	583.48 ± 244.77	706.20 ± 263.56	0.002 *
LF (ms^2^)	405.84 ± 173.17	448.89 ± 171.33	0.013 *
HF (ms^2^)	167.63 ± 77.57	185.69 ± 72.03	0.017 *
LF/HF ratio	5.18 ± 2.83	5.85 ± 2.79	0.018 *
Time domain			
SDNN (ms)	96.26 ± 30.66	98.25 ± 28.31	0.504
SDANN (ms)	96.79 ± 31.62	99.79 ± 31.17	0.278
ASDNN (ms)	49.18 ± 12.48	50.26 ± 11.16	0.366
rMSSD (ms)	25.89 ± 6.95	26.85 ± 6.83	0.164
pNN50 (%)	48.51 ± 24.97	49.38 ± 22.96	0.719

Values are presented as the mean ± standard deviation. * *p* < 0.05. HRV, heart rate variability; VLF, very low frequency; LF, low frequency; HF, high frequency; SDNN, standard deviation of intervals of all normal beats; SDANN, standard deviation of five-minute mean R-R interval; ASDNN, mean of five-minute standard deviations of intervals; rMSSD, root mean square of the difference of successive R-R intervals; pNN50, percentage of intervals that were more than 50 ms different from the previous interval.

**Table 5 nutrients-14-05320-t005:** Correlation analysis of serum albumin, prealbumin, and transferrin levels with the HRV parameters.

	Albumin	Prealbumin	Transferrin
Frequency domain			
VLF (ms^2^)	*r* = 0.314	*r* = 0.313	*r* = 0.307
	*p* = 0.003 *	*p* = 0.004 *	*p* = 0.009 *
LF (ms^2^)	*r* = 0.223	*r* = 0.226	*r* = 0.219
	*p* = 0.003 *	*p* = 0.002 *	*p* = 0.003 *
HF (ms^2^)	*r* = 0.218	*r* = 0.257	*r* = 0.254
	*p* = 0.005 *	*p* = 0.004 *	*p* = 0.005 *
LF/HF ratio	*r* = 0.256	*r* = 0.257	*r* = 0.254
	*p* = 0.008 *	*p* = 0.007 *	*p* = 0.001 *
Time domain			
SDNN (ms)	*r* = 0.124	*r* = 0.127	*r* = 0.126
	*p* = 0.011 *	*p* = 0.009 *	*p* = 0.009 *
SDANN (ms)	*r* = 0.099	*r* = 0.116	*r* = 0.122
	*p* = 0.041 *	*p* = 0.030 *	*p* = 0.034 *
ASDNN (ms)	*r* = 0.116	*r* = 0.122	*r* = 0.121
	*p* = 0.017 *	*p* = 0.012 *	*p* = 0.012 *
rMSSD (ms)	*r* = 0.122	*r* = 0.127	*r* = 0.121
	*p* = 0.011 *	*p* = 0.009 *	*p* = 0.012 *
pNN50 (%)	*r* = 0.075	*r* = 0.079	*r* = 0.076
	*p* = 0.122	*p* = 0.104	*p* = 0.118

* *p* < 0.05. HRV, heart rate variability; VLF, very low frequency; LF, low frequency; HF, high frequency; SDNN, standard deviation of intervals of all normal beats; SDANN, standard deviation of five-minute mean R-R interval; ASDNN, mean of five-minute standard deviations of intervals; rMSSD, root mean square of the difference of successive R-R intervals; pNN50, percentage of intervals that were more than 50 ms different from the previous interval.

**Table 6 nutrients-14-05320-t006:** Multiple regression analysis between nutritional markers and HRV parameters.

		Standardized β	B	*p*-Value	Adjusted R^2^
Albumin	Constant		0.284		0.174
	VLF	0.237	0.005	<0.001 **	
	LF/HF ratio	0.157	0.032	0.001 *	
	LF	0.139	0.005	0.002 *	
	HF	0.128	0.010	0.005 *	
Prealbumin	Constant		11.040		0.176
	VLF	0.236	0.003	<0.001 **	
	LF/HF ratio	0.157	0.207	0.001 *	
	LF	0.143	0.003	0.002 *	
	HF	0.130	0.006	0.005 *	
Transferrin	Constant		40.408		0.170
	VLF	0.230	0.089	<0.001 **	
	LF/HF ratio	0.156	5.485	0.001 *	
	LF	0.140	0.081	0.002 *	
	HF	0.131	0.171	0.005 *	

Variables are based on their order of listing in multiple regression analysis. * *p* < 0.05, ** *p* < 0.001. HRV, heart rate variability; B, regression coefficient; VLF, very low frequency; LF, low frequency; HF, high frequency; SDNN, standard deviation of intervals of all normal beats; SDANN, standard deviation of five-minute mean R-R interval; ASDNN, mean of five-minute standard deviations of intervals; rMSSD, root mean square of the difference of successive R-R intervals; pNN50, percentage of intervals that were more than 50 ms different from the previous interval.

## Data Availability

The datasets generated and/or analyzed during the current study are available from the corresponding author on reasonable request.

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
