# Peer review of "Nutritional Biomarkers and Heart Rate Variability in Patients with Subacute Stroke"

_nutrients, 2022, doi:10.3390/nu14245320_

Round 1
Reviewer 1 Report
The role of serum biomarkers in the diagnosis or monitoring of malnutrition is rather controversial. This is due to their relatively low specificity and the fact that major diseases, such as inflammation, are highly influenced, especially by visceral serum proteins.
In addition, the role of biomarkers in the management of nutritional therapy has not been studied in large randomized controlled trials. That said, many studies have shown that biomarkers such as prealbumin are reliable prognostic indicators of disease outcome and mortality in patients at risk for malnutrition.
In this regard, the presented results of a 12-year retrospective study are relevant and an interesting addition to the knowledge of the relationship of malnutrition biomarker levels with changes in HRV parameters in stroke patients. despite some limitations mentioned by the authors.
The manuscript submitted for review is clear, presented in a well-structured form, and is scientifically sound. The research methods are described in detail and clearly. The data obtained in the study are clearly reflected in tables. The interpretation of the data obtained is presented clearly and consistently throughout the manuscript.
The statistical analysis of the data is up-to-date, sound, and understandable.
Nevertheless, I have a few comments:
1. In my opinion, the results presented in the work don't correspond to the stated aim of the study- the work describes the results of studying the relationship between nutritional biomarkers and changes in HRV parameters, rather than the relationship of malnutrition with impaired ANS function in patients with stroke. It is necessary to adjust the formulation of the study objective.
2. Cited references need to be updated, since only 6 of 38 are recent (in the last 5 years).
Author Response
Please see the attachment.
Thank you again for your valuable reviews and suggestions.

Reviewer 2 Report
I have read the manuscript with interest. Regarding the introduction, it is perfect and needs no changes as it introduces the topic and properly defines the use of HRV as a useful tool for the given topic.
Material and method section is well writen and provides all the information needed ro understand the procedures.
There is a mistake in the names of the authors, it appears as there is an author called MD.
L113. Change "y" for "years"
The table 6 shows a multiple regression analysis. As the equation retrieved from the regression must be [y=m1x1 + m2x2 + m3x3....+ n]. Why aren't authors swowing the value of n?
The discussion section is the weakest of the paper. As the authors showed those many tables, there should be independent paragraphs concerning the relationship between HRV and Transferrin, Prealbumin...
Author Response

(The authors gave the same response as above.)
